# Toxicological Screening of Four Bioactive Citroflavonoids: In Vitro, In Vivo, and In Silico Approaches

**DOI:** 10.3390/molecules25245959

**Published:** 2020-12-16

**Authors:** Rolffy Ortiz-Andrade, Jesús Alfredo Araujo-León, Amanda Sánchez-Recillas, Gabriel Navarrete-Vazquez, Avel Adolfo González-Sánchez, Sergio Hidalgo-Figueroa, Ángel Josabad Alonso-Castro, Irma Aranda-González, Emanuel Hernández-Núñez, Tania Isolina Coral-Martínez, Juan Carlos Sánchez-Salgado, Victor Yáñez-Pérez, M. A. Lucio-Garcia

**Affiliations:** 1Laboratorio de Farmacología, Facultad de Química, Universidad Autónoma de Yucatán, Merida 97069, Yucatan, Mexico; amanda.sanchez@correo.uady.mx (A.S.-R.); victor.yanez@anahuac.mx (V.Y.-P.); 2Grupo de Investigación en Química Analítica y Ambiental, Facultad de Química, Universidad Autónoma de Yucatán, Merida 97069, Yucatan, Mexico; jesus.araujo@correo.uady.mx (J.A.A.-L.); tcoral@correo.uady.mx (T.I.C.-M.); monica.lucio@correo.uady.mx (M.A.L.-G.); 3Facultad de Química, Unidad Académica de Ciencias y Tecnología de la UNAM en Yucatán, Sierra Papacal 97302, Yucatan, Mexico; 4Laboratorio de Química Farmacéutica, Facultad de Farmacia, Universidad Autónoma del Estado de Morelos, Cuernavaca 62209, Morelos, Mexico; gabriel_navarrete@uaem.mx; 5Facultad de Ingeniería, Universidad Autónoma de Yucatán, Merida 92203, Yucatan, Mexico; avel.gonzalez@correo.uady.mx; 6Research, Innovation and Development Consortium for Arid Areas, Potosino Institute of Scientific and Technological Research, San Luis Potosi 78216, San Luis Potosi, Mexico; sergio.hidalgo@ipicyt.edu.mx; 7División de Ciencias Naturales y Exactas, Universidad de Guanajuato, Guanajuato 36050, Guanajuato, Mexico; angeljosabad@hotmail.com; 8Licenciatura en Nutrición, Facultad de Medicina, Universidad Autónoma de Yucatán, Merida 97000, Yucatan, Mexico; irma.aranda@correo.uady.mx; 9Departamento de Recursos del Mar, Centro de Investigación y de Estudios Avanzados del Istituto Politécnico Nacional-Unidad Mérida, Merida 97205, Yucatan, Mexico; emanuel.hernandez@cinvestav.mx; 10Hypermedic MX, Hacienda Santa Cecilia 97, Cafetales I, Coyoacán 04930, Mexico City, Mexico; juanc.sanchez@hypermedic.com.mx; 11Bioterio de la Escuela de Medicina, Universidad Anáhuac-Mayab, Merida 97310, Yucatan, Mexico

**Keywords:** citroflavonoids, low-risk substances, acute oral toxicity, MTT-based assay, toxicity prediction

## Abstract

Many studies describe different pharmacological effects of flavonoids on experimental animals and humans. Nevertheless, few ones are confirming the safety of these compounds for therapeutic purposes. This study aimed to investigate the preclinical safety of naringenin, naringin, hesperidin, and quercetin by in vivo, in vitro, and in silico approaches. For this, an MTT-based cytotoxicity assay in VERO and MDCK cell lines was performed. In addition, acute toxicity was evaluated on Wistar rats by OECD Guidelines for the Testing of Chemicals (Test No. 423: Acute Oral Toxicity-Class Method). Furthermore, we used the ACD/Tox Suite to predict toxicological parameters such as hERG channel blockade, CYP450 inhibition, and acute toxicity in animals. The results showed that quercetin was slightly more cytotoxic on cell lines (IC_50_ of 219.44 ± 7.22 mM and 465.41 ± 7.44 mM, respectively) than the other citroflavonoids. All flavonoids exhibited an LD_50_ value > 2000 mg/kg, which classifies them as low-risk substances as OECD guidelines established. Similarly, predicted LD^50^ was LD_50_ > 300 to 2000 mg/kg for all flavonoids as acute toxicity assay estimated. Data suggests that all these flavonoids did not show significant toxicological effects, and they were classified as low-risk, useful substances for drug development.

## 1. Introduction

Ethnomedicine is a millenary practice on bioactive compounds from plants and animals useful for therapeutics. According to the World Health Organization (WHO), medicinal plants are underlying sources of many medicines, so they should be thoroughly investigated to demonstrate their safety and efficacy on humans [1]. In this context, one of the most studied medicinal plants is *Citrus sp*., which has prominent medicinal properties, such as digestive, antiseptic, diuretic, antibacterial, antiviral, and beneficial effects on cancer, cardiovascular disease (CVD), and diabetes [2]. *Citrus sinensis* (sweet orange) is the most important and widely spread *Citrus sp*. around the world [1]. Some studies have shown that daily consumption of orange preparations reduces the prevalence of CVD risk factors. The main bioactive compounds responsible for this effect are flavonoids [1,2,3,4].

Citrus flavonoids (citroflavonoids) are promising agents that have been widely studied for several diseases. These phytochemicals have shown interesting biological activity in diabetes and its complications [5], cardiovascular disease [6], as well as virus and bacterial infections [7,8]. In addition, these compounds have shown antioxidant [9], estrogenic [10], and anticancer [11] activity. Despite their widely drug-like features, the safety profile of these molecules has not been comprehensively investigated.

There is toxicological evidence on animals, in vitro preparations, and by in silico predictions that reveal the safety of citroflavonoids. Lopes-Andrade et al. reported safety parameters of agathisflavone, a biflavonoid composed of two units of apigenin. These parameters were half lethal dose and hematological, biochemical, histopathological, behavioral, and physiological changes produced by the oral administration of this compound [12]. In addition, Lee et al. reported the repeated-dose of 3-months and 6-months oral toxicity of naringin on SD rats [13,14] and beagle dogs [15]. Of the flavonoid compounds reported in literature, quercetin exhibited relatively more cytotoxic effects than other flavonoids, but the evidence is controversial. Similar toxicity data are lacking for these flavonoids [16].

Early evidence reveals that citroflavonoids isolated from *C. sinensis* have significant hypoglycemic and antidiabetic effects [5]. Despite evidence about the efficacy of flavonoids in experimental models, there is insufficient information on safety parameters. Since the safety of herbal preparations use is one strategic objective established by WHO, we aimed to evaluate the toxicological potential of these citroflavonoids by a toxicological screening composed of in vitro, in vivo, and in silico assays.

## 2. Results

### 2.1. Cytotoxicity

The cytotoxicity assay was performed on VERO and MDCK cells using the MTT-based reduction method. Only QRT (quercetin) produced a decrease in the percentage viability of VERO cells at 250–750 µg/mL (IC_50_ = 465.41 µM or 140.66 µg/mL). In addition, for MDCK cells, QRT caused significant differences in the rate of cell viability at concentrations of 250 µg/mL and 500 µg/mL. On the other hand, HESP (hesperidin), NARGE (naringenin), and NAR (naringin) did not change the percentage of growth of both cells tested at concentrations ranging from 125 to 750 µg/mL (Figure 1).

### 2.2. Acute Oral Toxicity

#### 2.2.1. Mortality and Clinical Signs

There were no deaths recorded in any group during the observation period, and there were no noxious signs associated with administration of different doses of flavonoids. We did not observe any gross and pathological findings in rats’ organs and tissues at necropsy (data not shown) for any doses tested. These results indicate that the oral LD_50_ of all flavonoids in rats was more significant than 2000 mg/kg of body weight.

#### 2.2.2. Body Weight

Additionally, there were no significant differences in weight gain between the QRT group and the control group (Figure 2). However, there were considerable weight gain differences on days 7 and 10 related to HESP administration at 300 mg/kg dose. On the other hand, as we can see in Figure 2, there is evidence that NARGE modifies rats’ weight at 300 and 2000 mg/kg dose during test days 2, 3, 9, and 10. Furthermore, NAR changed the body weight of experimental animals.

#### 2.2.3. Hematological and Biochemical Analysis

The hematological parameters for all flavonoids-treated and control rats are shown in the Appendix A. The erythrocyte values at a dose of 50 mg/kg QRT and mean cell hemoglobin concentration (MCHC) values at 50 mg/kg and 300 mg/kg QRT were significantly lower than that of the control (Appendix A). The neutrophils count at a dose of 300 mg/kg, and 2000 mg/kg HESP were markedly lower than the control group (Appendix A). Regarding NARGE, only platelet values were modified at a dose of 2000 mg/kg and were significantly higher than that of the control (Appendix A). Finally, MCHC values at a 300 mg/kg NAR dose were considerably higher than that of the control (Appendix A). Other hematological parameters were not changed after 14 days of treatment (*p* > 0.05).

On the other hand, biochemical parameters for all flavonoids-treated and control group rats are shown in Table 1. Concerning glucose metabolism HESP, NARGE, and NAR, slightly amended glucose values. This was more evident at NAR 300 mg/kg and 2000 mg/kg, and NARGE 300 mg/kg. ALT and AST values in almost all flavonoids groups were not significantly higher than that of the control groups. Conversely, QRT (50 mg/kg and 300 mg/kg) induced that ALT and AST values were significantly higher than control. IB, DB, and TB values were not modified compared to the control group. However, QRT (50 mg/kg) caused an increase in DB values compared to the control. Finally, ALB, TP, and GL values were not modified compared to the control group after 14 days of treatment with each flavonoids tested (Table 2).

### 2.3. In Silico Prediction of Toxicity

#### 2.3.1. hERG Channel Blockade

The prediction outcomes of hERG channel blockade showed that QRT and NARGE have a low probability of generating cardiotoxic effects at clinically relevant concentrations (<10 mM). In contrast, for HESP and NAR, the probability is null (Table 3).

#### 2.3.2. Inhibition of CYP 450

The prediction results of inhibition of CYP 450 showed that, according to their molecular structure, QRT has a high probability of behaving as an inhibitor of isoform 1A2 of CYP 450 at clinically relevant concentrations (10 mM and 50 mM). On the other hand, NARGE has a high probability of behaving as an inhibitor of CYP 450, especially 2C9, 2C19, and 1A2 isoforms at clinically relevant concentrations and, to a lesser extent, inhibit 3A4 isoform. Finally, HESP and NAR have very low probabilities of behaving like inhibitors of CYP 450 at clinically relevant concentrations (Table 3).

#### 2.3.3. In Silico Acute Toxicity

We can see that predictive studies classified to QRT, HESP, and NARGE as members of category 4 of toxicity (2000 mg/kg < LD_50_ > 300 mg/kg) and NAR as member of category 3 or 4 (300 mg/kg < LD_50_ > 50 mg/kg or 2000 mg/kg < LD_50_ > 300 mg/kg) according to guide 423 of OECD [17] (Table 3).

## 3. Discussion

Herbal preparations have a role in modern medicine and the process of discovering, designing, and developing new drugs. Thus, there is clear evidence of their therapeutic benefits [18]. The common use of herbal products or isolated compounds has provided a strong rationale for investigating short and long-term toxicological effects [19]. It is essential to state that a thorough evaluation of the risk/benefit ratio is crucial to drug discovery in a limited efficacy evidence scenario and established toxicity. Therefore, it is important to evaluate cytotoxicity and short- and long-term toxicological profiles.

This study evaluated the cytotoxic activity of QRT, HESP, NARGE, and NAR against VERO and MDCK cell lines by MMT assay. As shown above, only QRT (250 µg/mL to 750 µg/mL) decreased VERO cell viability. Now, regarding MDCK cells, QRT caused significant differences in the percentage of cell viability at concentrations of 250 µg/mL and 500 µg/mL. In this context, QRT should be categorized as a moderate toxicity compound based on NCI. However, HESP, NARGE, and NAR did not show changes in the percentage of growth of both cell lines, so they could be categorized as low toxicity [20]. QRT biological effects possibly come from its antioxidant properties. However, this flavonoid may cause slightly toxic effects due to its concentration-dependent antioxidant and pro-oxidant properties, since it may act as a cytotoxic agent after its metabolic activation to produce semiquinone and quinoidal scaffolds. Therefore, byproduct formation must be considered in drug design ever [21]. Boncler et al. reported that NAR was not cytotoxic in a panel of five cell lines (5–50 µg/mL) since it did not modify mitochondrial membrane potential, cell membrane integrity, and nuclear morphology. In addition, this group showed that NAR was the lesser toxic compound in all the assays than other comparators [22].

Acute toxicity study showed that oral administration of flavonoids ranging from 50 mg/kg to 2000 mg/kg did not produce mortality, nor signs nor symptoms of apparent toxicity in rats. There were no gross abnormalities or pathological alterations at the end of the 14 days. This evidence is consistent with early report where acute oral LD_50_ for HESP was calculated as 4837.5 mg/kg [23]. This is larger than 2000 mg/kg, so it would be relatively safe as OECD guide 423 dictates. There was no significant effect observed in the body weight of rats treated with QRT. However, there were substantial weight gain differences on days 7 and 10 associated with HESP administration (300 mg/kg). Li reported that sub-chronic oral administration of HESP for 13 weeks did not modify weight gain at 250 and 500 mg/kg in Sprague-Dawley rats, but did at 1000 mg/kg. In addition, at last dose hemoglobin concentration, erythrocytes, and leukocytes count were increased [22]. In addition, NARGE and NAR modified rat weights at 300 and 2000 mg/kg. These fluctuations were probably due to slight decrease of food and water consumption. Knapp reported that food and water intake should be associated with CNS function and anorexia [24].

The hematological analysis showed relevant modifications. For instance, a significant decrease of erythrocyte count was observed in the QRT 50 mg/kg group. Furthermore, MCHC was decreased in both QRT groups. These findings indicated a possibility of anemia development. Kelly and other authors propose a non-significant effect on hemoglobin, red blood cell, and packed cell volume in these conditions [25,26,27]. In addition, MCHC is a biochemical parameter that correlates mean concentration of Hb in erythrocytes [28]. In our study, this parameter was lower in the QRT group than the control group, even when Hb was normal. In addition, neutrophil number was significantly lower than the control group at HESP 300 mg/kg and 2000 mg/kg. This indicates that this flavonoid causes short-term agranulocytosis. It is noteworthy that agranulocytosis is a severe blood disorder with a mortality rate of 3% to 8%. However, no previous reports had been published on HESP- induced agranulocytosis [29].

On the other hand, our study showed evidence of an increased number of platelets when NARGE at 2000 mg/kg was administered. Although these results could indicate a potential liver damage by NARGE administration, PT and aPTT tests reveal normal values, indicating low probability of this adverse condition. Therefore, this suggests that flavonoids did not induce anemia. Nyska and colleagues found no significant difference in the number of erythrocytes or MCHC. They also showed a decrease in eosinophils in Sprague-Dawley rats fed for 90 days with 5% alpha-glycosyl isoquercitrin [30].

HESP, NARGE, and NAR also slightly modified glucose. Some reports have mentioned that flavonoids used in this study decrease glucose levels as a common biological effect [31,32]. Thus, HESP and NAR produced biological activity by enhancing the antioxidant defense system and suppressing the production of proinflammatory cytokines [33]. Therefore, results suggest that QRT, HESP, NARGE, and NAR did not cause a toxic effect on carbohydrate metabolism. Interestingly, ALT and AST in all flavonoids groups were not significantly higher than control groups. Only QRT (50 mg/kg and 300 mg/kg) groups produced an increase of liver enzymes than the control. It has been well defined in literature that when transaminases increase, a liver toxicity process is occurring [34,35,36]. However, it should be point that ALT is a blood marker that also could be altered in skeletal muscle, heart, adipose tissue, and gut [37].

León and colleagues reported biochemical parameters of Sprague Dawley rats. This study showed that normal AST in these animals was 42.9–270.8 U/L, while ALT was 33.4–73.1 U/L [38]. When levels of transaminases (ALT and AST) are modified in pathological conditions a 10-fold increase takes place. Hepatitis and toxic-induced liver disease are examples where this type of alterations happen [39]. Another study has exposed that Sprague-Dawley rats have an increase of ALT, gamma-glutamyl transpeptidase (GGT), and bile acid when fed with 5% alpha-glycosyl isoquercitrin for 90 days. However, the results were found only in females [30]. Furthermore, increased alkaline phosphatase (ALP) has been reported when 128 or 450 mg/kg of quercetin was orally received for 14 days. In fact, bilirubin also increased, but only at 128 mg/kg. AST declined when a higher dose was administered [40]. Nyska et al. also found increases in the relative weight of kidneys in males, and the liver in both sex. No microscopic changes were found [30]. As mentioned above, the increase in ALT and bilirubin in the QRT group could be due to acute liver damage during compound administration.

Despite ALT values in QRT 50 mg/kg group not being 10-fold higher than the control group, there is a trend to increase this parameter after 14 days. The hepatotoxicity described above also could be associated with an increased TB or could be involved in further biliary excretion illnesses [41]. Taking into account that QRT 50 mg/kg caused ALT, AST, and DB alterations in a dose-independent manner, further investigation is mandatory for verifying safety of this flavonoid.

QSAR analysis is vital nowadays because it is one of the most economic, time saving and reliable tool for toxicological predictions in drug discovery. The ACD/Tox suite uses a QSAR-based algorithm that predicts toxicological effects in hERG channels and CYP450, as well as some parameters of genotoxicity, acute toxicity, aquatic toxicity, health alterations, irritation, and endocrine system disruption [42,43]. According to the results, QRT and NARGE have a low probability for inducing cardiotoxicity by hERG blocked at a clinically relevant concentration (<10 mM). Interestingly, HESP and NAR had minor probability.

Inhibition of CYP450 isoenzymes is also a clinically important drug interaction in toxicokinetic profile of bioactive substances. One of these inhibitors is grapefruit juice, a beverage with potent inhibitory activity of 3A4 isoenzyme. In addition, echinacea extract produces a weak inhibition of IA2 and 3A4 isoenzymes. Additionally, *Ginkgo biloba* is considered a weak inhibitor of 3A4 [44]. According to the results, QRT had high probability to inhibit isoform 1A2 at clinically relevant concentrations (10 mM and 50 mM). Furthermore, NARGE also showed a high probability to inhibit 2C9, 2C19, and 1A2 isoforms, and to a lesser extent, the 3A4 isoform. HESP and NAR had the lowest to inhibit CYP450.

Finally, predictive analyzes classified QRT, HESP and NARGE into category 4 (2000 mg/kg < LD_50_ > 300 mg/kg) of the OECD guide 423. NAR was classified into category 3 or 4 (300 mg/kg < LD_50_ > 50 mg/kg or 2000 mg/kg < LD_50_ > 300 mg/kg). It is very important to note that predictive outcomes were similar than those obtained in experimental models. Prediction indicated an oral LD_50_ higher than 2000 mg/kg for all flavonoids evaluated, so they were classified into category 4.

In conclusion, QRT, HESP, NARGE, and NAR are potential bioactive compounds with a good safety profile based on NCI criteria. In this context, they might be categorized as low-toxicity agents. Furthermore, these compounds were relatively safe based on OECD guide 423.

## 4. Materials and Methods

### 4.1. Chemicals and Reagents

Quercetin (QRT), hesperidin (HSP), naringenin (NARGE), and naringin (NAR), dimethyl sulfoxide (DMSO), tetrazolium dye (MTT), and cisplatin were purchased from Sigma–Aldrich Co. (St. Louis, MO, USA). All other reagents were analytical grade from local sources.

### 4.2. Cell Cultures

VERO and MDCK cell lines were acquired from American Type Culture Collection (ATCC, Manassas, VA, USA). Cells were seeded in Advanced Dulbecco’s modified Eagle’s medium (DMEM), supplemented with 10% inactivated newborn calf serum, antibiotics (100 U/mL of penicillin, and 100 mg/mL of streptomycin), and incubated at 37 °C in a humidified atmosphere of 5% CO_2_.

### 4.3. Animals

Female Wistar rats (250 g b.w.) were used. These animals were obtained from Bioterium from División de Ciencias de la Salud at the Universidad de Quintana Roo. They were maintained under standard environmental conditions (22 ± 3 °C, 12 h/12 h light/dark cycle), and free access to standard rodent diet and water ad libitum. All animal procedures were conducted under our Federal Regulations for Animal Experimentation and Care [45] and were approved by the Institutional Animal Care and Use Committee of the “Universidad Juárez Autónoma de Tabasco” university (Code:2017-001, Approved: October 2017). Rats were acclimatized for 1 week before the start of experimental procedures.

### 4.4. MTT-Based Cytotoxicity Assay

In vitro cytotoxicity assay was performed on Madin-Darby Canine Kidney Epithelial Cells (MDCK) and green Monkey Kidney Cells (VERO) by MTT-based assay. Briefly, cells were seeded in 96-well plates at a density of 2.5 × 10^3^ cells/well. Following 24 h of incubation at 37 °C with 5% CO_2_, the cells were treated with varying concentrations of samples (1000, 750, 500, 250, and 125 µg/mL) for 48 h for quadruplicate. Following washing and incubation with MTT solution (1 mg/mL) for 4 h, the cells were lysed. The absorbance was measured after 45 min using a microplate reader (Multiskan Asent^®^) at a wavelength of 550 nm.

The criteria for categorizing cytotoxicity against MDCK and VERO cells are based on the U.S. National Cancer Institute (NCI). The criteria comprise as follow: IC_50_ ≤ 20 μg/mL = highly cytotoxic, IC_50_ ranged between 21 and 200 μg/mL = moderately cytotoxic, IC_50_ ranged between 201 and 500 μg/mL = weakly cytotoxic and IC_50_ > 501 μg/mL = no cytotoxicity [9].

### 4.5. Acute Toxicity Test

The half lethal dose (LD_50_) was estimated using the fixed-dose method described in OECD Guidelines for the Testing of Chemicals (Test No. 423: Acute Oral Toxicity-Class Method; OECD, 2001). Briefly, four fixed-doses (5, 50, 300, and 2000 mg/kg) were used by a standardized method LD_50_. This method allowed a step-by-step assay according to the chosen dose to begin the experiment. Subsequently, depending on the presence or absence of mortality, the decision to continue within the next step is taken, so fewer experimental animals should be used. For purposes of this study, QRT at 50 and 300 mg/kg, as well as HESPE, NARGE and NAR at 300 and 2000 mg/kg were orally administered.

Rats used for this experiment were fasted for 8 h and were organized into two groups of three rats for each dose and each flavonoid. Likewise, the control group received isotonic 0.9% sodium chloride solution (dose: 5 mL/kg). All rats were closely observed for toxic symptoms and behavioral changes (sedation, hyperactivity, writhing, pruritus, feeding and drinking habits, general morphological changes, and bite legs) for 30 min and periodically 24 h post-administration. Animals were observed daily (14 days) and mortality rate recorded. A LD_50_ was estimated according to criteria of OECD guide 423.

The rats’ body weight in each group was determined and recorded before the treatment initiation and after 14 days. Changes in the body weight at the end of the 14 days were calculated. Once observation time was completed and to detect any adverse effect, blood samples were collected by cardiac puncture after the previous fasting of 8 h. For this purpose, rats were anesthetized by intraperitoneally 50 mg/kg sodium pentobarbital. The liver was carefully extracted, rinsed in normal saline, drained on blotting paper, and carefully examined for tissue lesions. In addition, organ weight was determined.

### 4.6. Hematological Analysis

Rat blood samples were analyzed by a hematology auto-analyzer system (Sysmex XS-1000i^®^, SYSMEX^®^, Oak Ridge, TN, USA). The parameters quantified on this stage were: blood count-packed cell volume (PCV), red blood cell (RBC) count, hemoglobin (Hb), platelet count, white blood cell (WBC) count, mean cell hemoglobin concentration (MCHC), mean red cell volume (MCV), mean cell hemoglobin (MCH), prothrombin time (PT), and activated partial thromboplastin time (aPTT).

### 4.7. Biochemical Analysis

Albumin (ALB), total protein (TP), glucose (GLU), aspartate transaminase (AST), alanine transaminase (ALT), indirect bilirubin (IB), direct bilirubin (DB), total bilirubin (TB), and globulin (GL) were quantified by standardized methods on the Automated Clinical System (COBAS INTEGRA^®^ 400 plus, ROCHE^®^).

### 4.8. In Silico Prediction of Toxicity

For this study, the ACD/Tox Suite^®^ (Advance Chemistry Development, Inc. Toronto, Canada) platform was used to predict the toxicity of QRT, HESP, NARGE, and NAR. This software uses a modeling QSAR algorithm based on Physiologically Based Pharmacokinetic Modelling. This platform is a specialized model that predicts hERG (the human Ether-à-go-go-Related Gene) blockage, CYP3A4 inhibition, genotoxicity assessment, acute toxicity (rodent LD_50_), aquatic toxicity, health effects, irritation, and endocrine system disruption [42,43]. Therefore, to carry out the prediction of baseline toxicity, molecules of each flavonoid were drawn by ACD ChemSketch. Subsequently, molecules were placed in ACD/Tox Suite^®^, and modules employed were acute toxicity group, LD_50_, hERG, and CYP450 inhibition. LD_50_ was classified to each of flavonoids according to five categories established by OECD. On the other hand, the probability of selected CYP450 isoform (3A4, 2D6, 2C9, 2C19, 1A2) inhibition was calculated at 10 and 50 mM. Furthermore, the blocking probability of hERG channels was assessed at 10 mM.

### 4.9. Statistical Analysis

All results were expressed as mean ± standard error. The results from the in vivo and in vitro experiments were analyzed using software Origin^®^ 8.0. Statistical analysis of data was carried out by the same software, where analysis of variance (ANOVA) at a 5% level of significance and Tukey test was used. The IC_50_ (half inhibitory concentration) values were calculated in Origin software^®,^ (Northampton, MA, USA) and one-way ANOVA analysis was compared with cisplatin as the control.

## 5. Conclusions

QRT, HESP, NARGE, and NAR are safety drug candidates that could be proposed as active substances in drug development due to their toxicological profiles. Thus, all these molecules are classified as relatively safe and practically non-toxic for human use. This work represents a part of a project focused on demonstrating the tolerability and safety of flavonoids in other organs and systems. The resulting data supports the innocuity and safety profile of these citroflavonoids.

## Figures and Tables

**Figure 1 molecules-25-05959-f001:**
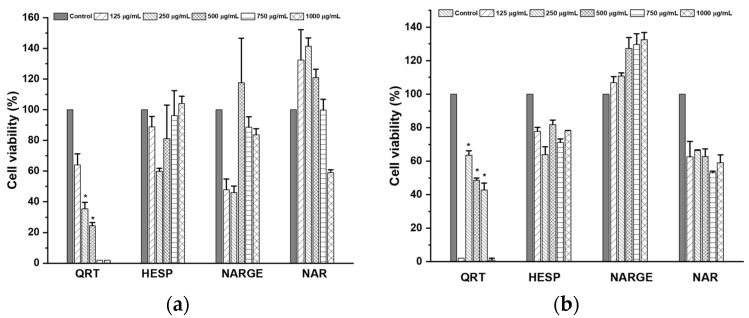
Percentage viability in (**a**) MDCK and (**b**) VERO cells after exposition with citroflavonoids. Non-tumorigenic cells were incubated with different concentrations of flavonoids for 48 h. Then, the MTT assay was performed. Values are presented as mean percent of viability ± standard error. *n* = 3, * *p* < 0.05.

**Figure 2 molecules-25-05959-f002:**
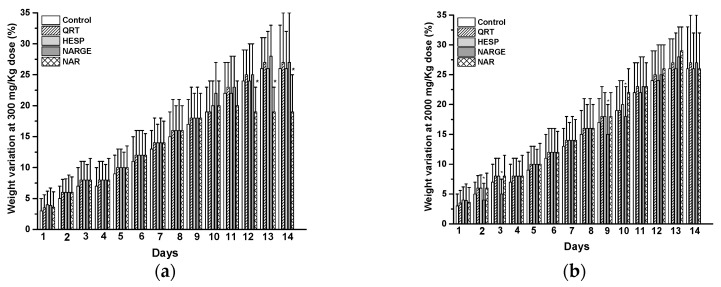
(**a**) Changes in body weight after treatment with control or quercetin (QRT), hesperidin (HESP), naringenin (NARGE), and naringin (NAR) at 300 mg/Kg dose, and (**b**) Changes in body weight after treatment with control or quercetin (QRT), hesperidin (HESP), naringenin (NARGE), and naringin (NAR) at 2000 mg/Kg dose. Results are presented as mean ± SE (*n* = 3), * *p* < 0.05.

**Table 1 molecules-25-05959-t001:** Biochemical analysis: values and effects after treatment with several doses of flavonoids.

Experimental Group	Dose (mg/kg)	Biochemical Marker					
		GLU (mg/dL)	AST (IU/L)	ALT (IU/L)	TB (mg/dL)	DB (mg/dL)	IB (mg/dL)
Control	-	114.50 ± 0.50	159.50 ± 13.82	58.00 ± 1.69	0.04 ± 0.002	0.010 ± 0.00	0.035 ± 0.002
QRT	50	119.67 ± 8.74	266.67 ± 32.29	83.00 ± 3.10 *	0.08 ± 0.006	0.043 ± 0.009 *	0.036 ± 0.003
QRT	300	149.49 ± 5.00	127.67 ± 4.12	64.00 ± 2.28 *	0.05 ± 0.003	0.016 ± 0.004	0.030 ± 0.006
Control	-	182.49 ± 5.70	123.95 ± 9.86	32.52 ± 6.02	0.057 ± 0.006	0.025 ± 0.006	0.031 ± 0.004
HESP	300	155.33 ± 18.71	126.00 ± 13.18	33.78 ± 1.30	0.051 ± 0.002	0.020 ± 0.005	0.029 ± 0.003
HESP	2000	192.06 ± 12.73	226.65 ± 44.80	41.19 ± 2.83	0.060 ± 0.06	0.018 ± 0.005	0.046 ± 0.010
Control	-	182.49 ± 5.70	123.95 ± 9.86	32.52 ± 6.02	0.057 ± 0.006	0.025 ± 0.006	0.031 ± 0.004
NARGE	300	145.29 ± 5.47 *	156.06 ± 9.33	39.19 ± 0.77	0.064 ± 0.005	0.013 ± 0.007	0.051 ± 0.007
NARGE	2000	175.56 ± 6.80	149.37 ± 7.16	57.68 ± 10.97	0.065 ± 0.007	0.020 ± 0.001	0.045 ± 0.060
Control	-	123.49 ± 5.70	123.95 ± 9.86	32.52 ± 6.02	0.057 ± 0.006	0.025 ± 0.006	0.031 ± 0.004
NAR	300	125.54 ± 14.59	144.58 ± 17.05	27.57 ± 0.62	0.037 ± 0.001	0.012 ± 0.0009	0.024 ± 0.0007
NAR	2000	138.55 ± 10.48	128.26 ± 5.33	38.59 ± 1.08	0.039 ± 0.004	0.015 ± 0.001	0.025 ± 0.003

Values are presented as the mean plasma concentration ± standard error. *n* = 3, * *p* < 0.05 with respect to control. GLU: glucose, AST: aspartate transaminase, ALT: alanine transaminase, TB: total bilirubin, DB: direct bilirubin, IB: indirect bilirubin.

**Table 2 molecules-25-05959-t002:** Biochemical analysis: values and effects after treatment with several doses of flavonoids.

Experimental Group	Dose (mg/kg)	Biochemical Marker			
		TP (g/dL)	ALB (g/dL)	GL (g/dL)	ALB/GL Ratio
Control	-	6.37 ± 0.03	3.85 ± 0.04	2.52 ± 0.00	1.52 ± 0.015
QRT	50	6.38 ± 0.15	3.88 ± 0.04	2.50 ± 0.14	1.56 ± 0.087
QRT	300	6.48 ± 0.065	3.63 ± 0.09	2.84 ± 0.16	1.39 ± 0.15
Control	-	6.41 ± 0.21	4.26 ± 0.23	2.14 ± 0.34	2.18 ± 0.44
HESP	300	5.91 ± 0.25	4.15 ± 0.20	1.76 ± 0.069	2.36 ± 0.101
HESP	2000	5.99 ± 0.11	4.20 ± 0.22	1.80 ± 0.108	2.38 ± 0.270
Control	-	6.41 ± 0.21	4.26 ± 0.23	2.14 ± 0.34	2.18 ± 0.44
NARGE	300	6.97 ± 0.30	3.86 ± 0.10	3.11 ± 0.25	1.26 ± 0.08
NARGE	2000	7.82 ± 0.07	4.74 ± 0.16	3.08 ± 0.17	1.56 ± 0.14
Control	-	6.41 ± 0.21	4.26 ± 0.23	2.14 ± 0.34	2.18 ± 0.44
NAR	300	6.48 ± 0.04	4.50 ± 0.02	1.97 ± 0.04	2.30 ± 0.05
NAR	2000	6.71 ± 0.14	4.66 ± 0.16	2.03 ± 0.05	2.32 ± 0.12

Values are presented as the mean plasma concentration ± standard error. *n* = 3, *p* < 0.05 with respect to control. TP: total protein, ALB: albumin, GL: globulin, ALB/GL: albumin/globulin ratio.

**Table 3 molecules-25-05959-t003:** In silico toxicity parameters calculated for all flavonoids.

**Inhibition of Different Isoforms of CYP450**						
**Flavonoid Compound**	**Concentration** **(µM)**	**3A4**	**2D6**	**2C9**	**2C19**	**1A2**
QRT	10	2	0	7	5	94
	50	83	23	68	37	95
HESP	10	1	1	4	1	0
	50	6	1	4	2	0
NARGE	10	3	1	51	82	67
	50	54	39	87	91	93
NAR	10	1	1	4	1	0
	50	4	1	4	1	0
**Blocking of hERG Channel**						
**Flavonoid Compound**	**Concentration** **(µM)**	**Probability** **(%)**				
QRT	10	14				
HESP	10	0				
NARGE	10	5				
NAR	10	0				
**Calculated Median Lethal Dose (LD_50_)**						
**Flavonoid Compound**	**Route of Administration**	**LD_50_** **(mg/kg)**				
QRT	Intraperitoneal	1200				
	Oral	640				
HESP	Intraperitoneal	1600				
	Oral	3000				
NARGE	Intraperitoneal	1200				
	Oral	1900				
NAR	Intraperitoneal	1500				
	Oral	1200				

Results were obtained with ACD/Tox Suite software version 2.95.

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
