# Peer review of "Toxicological Screening of Four Bioactive Citroflavonoids: In Vitro, In Vivo, and In Silico Approaches"

_molecules, 2020, doi:10.3390/molecules25245959_

Round 1

Reviewer 1 Report

The present article shows a toxicological screening of citrus flavonoids: quercetin, naringenin, naringin and hesperidin trough in vivo, in silico and in vitro approaches. The obtained results indicate that they are low-risk compounds except for quercetin that show a moderate risk.

The authors made some modifications suggest in a first evaluation as the correction of figures 1 and 2 and the changes presented along with the other alterations/additions are satisfactory and do not require major corrections to be made.

There are minor corrections trough the article as follows:

  • change µg/ml for µg/mL all over the text. (If doubts about the correct use of symbols, please check the green book of IUPAC)
  • page 6 lines 154-155: just put together the paragraphs, there is an additional “enter”.
  • page 6 line 162: I believe there is an additional “are” here: “…rich plant extracts are evaluated [20].”

Author Response

The present article shows a toxicological screening of citrus flavonoids: quercetin, naringenin, naringin and hesperidin trough in vivo, in silico and in vitro approaches. The obtained results indicate that they are low-risk compounds except for quercetin that show a moderate risk.

The authors made some modifications suggest in a first evaluation as the correction of figures 1 and 2 and the changes presented along with the other alterations/additions are satisfactory and do not require major corrections to be made.

There are minor corrections trough the article as follows:

  • change µg/ml for µg/mL all over the text. (If doubts about the correct use of symbols, please check the green book of IUPAC)
  • page 6 lines 154-155: just put together the paragraphs, there is an additional “enter”.
  • page 6 line 162: I believe there is an additional “are” here: “…rich plant extracts are evaluated [20].”

Answer.
The units format was corrected and the RESULTS and DISCUSSION sections were rewritten. The additional spaces in the text were also modified and two bibliographic references were included to support the discussion of the results.

Reviewer 2 Report

Manuscript number: molecules-1031270

Title: Toxicological screening of four bioactive citroflavonoids: in vitro, in vivo, and in silico approaches.

As I have mentioned before, I like the evaluate the toxicity of several bioactive citroflavonoids using in vitro, in vivo an in silico approaches. The numerous biological properties of phenolic compounds and flavonoids have been extensively reported, but it exists a lack of knowledge about their toxicity.

In the following paragraphs, I will provide clear information in order to improve the manuscript. In general, the English level is good and I cannot see many discrepancies.

INTRODUCTION

The introduction has been extended with relevant and clear information. However, I would encourage the authors to add more context. For example, why are controversial the evidences of quercetin’s toxicity? The final objective is clear and accurate.

  • Line 59: Citrus
  • Line 72-74: please, rephrase.

RESULTS

Authors have performed several changes in this section in other to improve it. Figure 1 has been modified and now is more understandable and accurate. However, regarding Figure 2, the different doses of the flavonoids have not been represented and only the significant differences in body weight caused by NAR are noted. I would recommend to revise both Figure and the section 2.2.2. Body weight.

DISCUSSION

The discussion of the results is wide, but I have several suggestions for the authors.

  • In some cases, in some cases, the authors mention studies to discuss their results in a slightly confusing way, in my opinion. For example, in Line 158-162, authors are presenting the results of a previous study, but the context is a bit unclear. I would recommend to rephrase the sentences, for example, adding “In a previous study…”.
  • Line 155:” only QRT decreased in the viability of…”.
  • Line 161: “of naringin is the lowest than resveratrol…”. Please, rephrase.
  • Lines 158-162.
  • Line 163, 171, 185: include the meaning of the abbreviature.
  • Line 192-200: references are missing.

FINAL CONCLUSIONS

From my point of view, authors have performed an interesting work. In this case, I am suggesting MINOR REVISIONS to improve the quality of this work.

Author Response

Title: Toxicological screening of four bioactive citroflavonoids: in vitroin vivo, and in silico approaches.

As I have mentioned before, I like the evaluate the toxicity of several bioactive citroflavonoids using in vitroin vivo an in silico approaches. The numerous biological properties of phenolic compounds and flavonoids have been extensively reported, but it exists a lack of knowledge about their toxicity.

In the following paragraphs, I will provide clear information in order to improve the manuscript. In general, the English level is good and I cannot see many discrepancies.

INTRODUCTION

The introduction has been extended with relevant and clear information. However, I would encourage the authors to add more context. For example, why are controversial the evidences of quercetin’s toxicity? The final objective is clear and accurate.

  • Line 59: Citrus
  • Line 72-74: please, rephrase.

Answer.
Adjustments were made in the introduction including information on the cause of the effects of quercetin toxicity related to the effects of its catechol group.

RESULTS

Authors have performed several changes in this section in other to improve it. Figure 1 has been modified and now is more understandable and accurate. However, regarding Figure 2, the different doses of the flavonoids have not been represented and only the significant differences in body weight caused by NAR are noted. I would recommend to revise both Figure and the section 2.2.2. Body weight.

Answer. Substantial changes were made in this section, modifying the wording. Figure 2 was modified, including the missing information.

DISCUSSION

The discussion of the results is wide, but I have several suggestions for the authors.

  • In some cases, in some cases, the authors mention studies to discuss their results in a slightly confusing way, in my opinion. For example, in Line 158-162, authors are presenting the results of a previous study, but the context is a bit unclear. I would recommend to rephrase the sentences, for example, adding “In a previous study…”.
  • Line 155:” only QRT decreased in the viability of…”.
  • Line 161: “of naringin is the lowest than resveratrol…”. Please, rephrase.
  • Lines 158-162.
  • Line 163, 171, 185: include the meaning of the abbreviature.
  • Line 192-200: references are missing.

Answer. All the recommendations have been addressed, rewriting the entire section and rearranging the way to approach the discussion of each flavonoid. Two more bibliographic references were added to support a better understanding of the section.

This manuscript is a resubmission of an earlier submission. The following is a list of the peer review reports and author responses from that submission.

Round 1

Reviewer 1 Report

The manuscript " Toxicological screening of four bioactive citroflavonoids: in vitro, in vivo, and in silico approaches" written by Rolffy Ortiz-Andrade and co-authors describes lack the toxic of four citroflavonoids.

The manuscript needs improvement before publishing.

The introduction to the thesis is very short and does not introduce in the researched problematic. On the other hand, the discussion is too long, as many places describe and discuss the results of the study that were not presented in the manuscript. Additionally, general issues are described (eg. ALT, ASPAT etc.).

The Material and Methods are generally described without essential details (e.g., administration of the solvent for test compounds), positive control, and how many replicates the MTT test was performed. In vivo tests also raise doubts in what form the substances were administered, etc.

The description of the results is not adequate to figure 1 and 2 presented. Lack of statistical determinations in Figures 1 and 2.  Lack of results from haematology, biochemistry and organ weights.

The conclusion from these studies is also highly controversial.

Author Response

  1. The manuscript " Toxicological screening of four bioactive citroflavonoids: in vitro, in vivo, and in silico approaches" written by Rolffy Ortiz-Andrade and co-authors describes lack the toxic of four citroflavonoids. Thank you for your comments.
  2. The manuscript needs improvement before publishing. Thank you for your comments. A comprehensive revision was made to a improved this manuscript.
  3. The introduction to the thesis is very short and does not introduce in the researched problematic. On the other hand, the discussion is too long, as many places describe and discuss the results of the study that were not presented in the manuscript. Additionally, general issues are described (eg. ALT, ASPAT etc.). Thank you for your comments. Modifications were made to the manuscript.
  4. The Material and Methods are generally described without essential details (e.g., administration of the solvent for test compounds), positive control, and how many replicates the MTT test was performed. In vivo tests also raise doubts in what form the substances were administered, etc. Thank you for your comments. Modifications were made to the manuscript.
  5. The description of the results is not adequate to figure 1 and 2 presented. Lack of statistical determinations in Figures 1 and 2. Lack of results from haematology, biochemistry and organ weights. Thank you for your comments. A comprehensive revision was made to a improved this manuscript. The corresponding changes were made. Thank you for your comments. Hematology results were annexed in supplementary tables called S1-S4.
  6. The conclusion from these studies is also highly controversial. Thank you for your comments. This section has been rewritten.

Reviewer 2 Report

The article it is about a toxicological screening of the flavonoid quercetin, naringenin, naringin and hesperidin trough in vivo, in silico and in vitro approaches. The results indicate that all are low-risk substances except for quercetin that show a moderate risk.

Abstract summarizes well the paper. There are some problems with the figures presented in the article, and some results need to be checked.

There are a few issues with English language. Minor corrections trough the article.

Follow some observations:

Introduction section must be improved. Page 2, line 62-63: must be complemented with information about the importance of the study in question. Why study the toxicity levels of these flavonoids?

Page 2 line 73: what HESP, NARGE, and NAR mean? It is described only in the material and methods section, ate the end of the article; it will be interesting and facilitate the reading if described it here.

Page 2 line 74: is not figure 1 but figure 2, the graphic reference for this paragraph. Check it.

Page 3, line 77-80: The figures are not complete. Check them. The legend for figure 1 is not consistent with the presented graphic. The same happen with figure 2.

Page 3 line 82: there is no reference to figure 2 in the text before or after this figure.

Page 3: Figure 1 and Figure 2 legends: change µg/ml for µg/mL.

Page 3 line 85: it will be interesting present all the collected data here, in the supplementary material.

Page 3, line 93; Page 4, lines 95 and 96: is not Figure 1 but Figure 2.

Page 4, line 99: MCHC? What means? Describe it.

Page 4: 97-105. The results here are from supplementary information, so, table 1 must be named as table S1 and so on.

Table 3: concentrations are given in M or mM? Please check it.

Page 6: line 133-134: check the information given here for NAR.

Page 6, line 146-147: Data not shown in figure 1.

Page 7, lines 149-151: the text here is confusing, rewrite it.

Page 7, line 152: need a reference here.

Page 8, line 227-229 – need a reference here.

Page 9 line 286-287: check this information about NAR with the results presented.

Page 9, line 291-293: rewrite, it is repetitive.

Page 10 line 312: check the units and subscribes.

Page 10, line 335: correct ml/Kg to mL/Kg.

Page 10, line 337: chance min for minutes, check the rest of the text.

Page 10, line 341: every 14 days or after 14 days?

Page 12: check the references. (1, 9, 18)

Author Response

  1. The article it is about a toxicological screening of the flavonoid quercetin, naringenin, naringin and hesperidin trough in vivo, in silico and in vitro approaches. The results indicate that all are low-risk substances except for quercetin that show a moderate risk. Thank you for your comments.
  2. Abstract summarizes well the paper. There are some problems with the figures presented in the article, and some results need to be checked. Thank you for your comments. Necessary changes have been made.
  3. There are a few issues with English language. Minor corrections trough the article. Thank you for your comments.

Follow some observations:

  1. Introduction section must be improved. Page 2, line 62-63: must be complemented with information about the importance of the study in question. Why study the toxicity levels of these flavonoids?
  2. Page 2 line 73: what HESP, NARGE, and NAR mean? It is described only in the material and methods section, ate the end of the article; it will be interesting and facilitate the reading if described it here.
  3. Page 2 line 74: is not figure 1 but figure 2, the graphic reference for this paragraph. Check it.
  4. Page 3, line 77-80: The figures are not complete. Check them. The legend for figure 1 is not consistent with the presented graphic. The same happen with figure 2.
  5. Page 3 line 82: there is no reference to figure 2 in the text before or after this figure.
  6. Page 3: Figure 1 and Figure 2 legends: change µg/ml for µg/mL.
  7. Page 3 line 85: it will be interesting present all the collected data here, in the supplementary material.
  8. Page 3, line 93; Page 4, lines 95 and 96: is not Figure 1 but Figure 2.
  9. Page 4, line 99: MCHC? What means? Describe it.
  10. Page 4: 97-105. The results here are from supplementary information, so, table 1 must be named as table S1 and so on.
  11. Table 3: concentrations are given in M or mM? Please check it.
  12. Page 6: line 133-134: check the information given here for NAR.
  13. Page 6, line 146-147: Data not shown in figure 1.
  14. Page 7, lines 149-151: the text here is confusing, rewrite it.
  15. Page 7, line 152: need a reference here.
  16. Page 8, line 227-229 – need a reference here.
  17. Page 9 line 286-287: check this information about NAR with the results presented.
  18. Page 9, line 291-293: rewrite, it is repetitive.
  19. Page 10 line 312: check the units and subscribes.
  20. Page 10, line 335: correct ml/Kg to mL/Kg.
  21. Page 10, line 337: chance min for minutes, check the rest of the text.
  22. Page 10, line 341: every 14 days or after 14 days?
  23. Page 12: check the references. (1, 9, 18).

Thank you for your comments. The request corrections will be made.

Reviewer 3 Report

Manuscipt: molecules-953370

Title: Toxicological screening of four bioactive citroflavonoids: in vitro, in vivo, and in silico approaches.

I like the effort of the authors to evaluate the pharmacological effects of several flavonoids, using different cell lines and model animals. In addition, some toxicological parameters have been predicted. In recent years, numerous studies have demonstrated that phenolic compounds, including many flavonoids, present bioactivities that could interesting for future industrial applications. However, in many cases, the toxicity has not been evaluated, so, in my opinion, this study is of great interest.

In general, the English level is good, the manuscript is clear and fluid and I cannot see many discrepancies. In the following paragraphs, I will provide clear information to improve the manuscript.

Introduction

In my opinion, the introduction is too short. More context and information should be explained. I would suggest to add more recent information about phenolic compounds and their possible industrial applications.

  • Line 54 and 57: The genus “Citrus” should be in italics (Citrus sp.)

Results

In this section, several abbreviations are used before their meaning is specified in the text, such as QRT, or HESP.

  • Figures 1 and 2. I do not see any (a) or (b) neither in Figure 1 nor 2. I would recommend to reformulate these Figures. In addition, in Figure 2, the results of 125 and 1000 µg/mL are missing.
  • In the section of body weight, the Figure 1 is mentioned, but this one is the percent of viability of the two cell lines. Maybe a figure is missing?
  • Tables 1 and 2 could be joined.
  • Regarding glucose values, it seems that HESP produced a higher increase than NAR.

Discussion

  • In the discussion, the authors mention that the toxicity of naringin has been assessed in different cell lines. Have no other similar studies been carried out with the other compounds?
  • What do the authors think the results of QRT might be due to?
  • Sometimes, the abbreviations are not used in the text, such as in Line 284. I would recommend to homogenize the manuscript.
  • Line 202: a comma should be used after MCHN.
  • Line 274: Ginko biloba should be in italics.

FINAL REMARKS

In my opinion, I am suggesting MAJOR REVISIONS before publishing. The authors have performed an interesting work but they should do some changes to improve its quality.

Author Response

  1. I like the effort of the authors to evaluate the pharmacological effects of several flavonoids, using different cell lines and model animals. In addition, some toxicological parameters have been predicted. In recent years, numerous studies have demonstrated that phenolic compounds, including many flavonoids, present bioactivities that could interesting for future industrial applications. However, in many cases, the toxicity has not been evaluated, so, in my opinion, this study is of great interest. Thank you for your comments.
  2. In general, the English level is good, the manuscript is clear and fluid and I cannot see many discrepancies. In the following paragraphs, I will provide clear information to improve the manuscript. Thank you for your comments. Necessary changes have been made.
  3. In my opinion, the introduction is too short. More context and information should be explained. I would suggest to add more recent information about phenolic compounds and their possible industrial applications. Thank you for your comments. Necessary changes have been made.
  • Line 54 and 57: The genus “Citrus” should be in italics (Citrus sp.). Necessary changes have been made.
  1. In this section, several abbreviations are used before their meaning is specified in the text, such as QRT, or HESP. Necessary changes have been made.
  • Figures 1 and 2. I do not see any (a) or (b) neither in Figure 1 nor 2. I would recommend to reformulate these Figures. In addition, in Figure 2, the results of 125 and 1000 µg/mL are missing. Necessary changes have been made.
  • In the section of body weight, the Figure 1 is mentioned, but this one is the percent of viability of the two cell lines. Maybe a figure is missing?
  • Tables 1 and 2 could be joined. Necessary changes have been made.

  1.  
  • In the discussion, the authors mention that the toxicity of naringin has been assessed in different cell lines. Have no other similar studies been carried out with the other compounds?
  • What do the authors think the results of QRT might be due to?
  • Sometimes, the abbreviations are not used in the text, such as in Line 284. I would recommend to homogenize the manuscript.
  • Line 202: a comma should be used after MCHN.
  • Line 274: Ginko biloba should be in italics.

Necessary changes have been made in this section.

Round 2

Reviewer 1 Report

Accept in present form

Reviewer 3 Report

Dear authors,

Please, could you be more specific in your comments. I believe you could have responded something else than "Necessary changes have been made". Honestly, if I am the editor I would have rejected the manuscript and forced you re-submit the manuscript. 

I don't have any other suggestion. What for should I waste my time on you? Go ahead and publish it as it is in its present form. It is your name, not mine the one that is signing the manuscript and its content. 

Sincerely,

A reviewer